# *MetaNovo*: An open-source pipeline for probabilistic peptide discovery in complex metaproteomic datasets

**Matthys G. Potgieter**[1,2], **Andrew J. M. Nel**[2], **Suereta Fortuin**[2], **Shaun Garnett**[2], **Jerome M. Wendoh**[3], **David L. Tabb**[2,4], **Nicola J. Mulder**[1,5]*, **Jonathan M. Blackburn**[2,5]*

1 Computational Biology Division, Department of Integrative Biomedical Sciences, University of Cape Town, Cape Town, South Africa, 2 Division of Chemical and Systems Biology, Department of Integrative Biomedical Sciences, University of Cape Town, Cape Town, South Africa, 3 Division of Immunology, Department of Pathology, University of Cape Town, Cape Town, South Africa, 4 Division of Molecular Biology and Human Genetics, Department of Biomedical Sciences; African Microbiome Institute; South African Tuberculosis Bioinformatics Initiative; Stellenbosch University, Cape Town, South Africa, 5 Institute of Infectious Disease & Molecular Medicine, Faculty of Health Sciences, University of Cape Town, Cape Town, South Africa

* nicola.mulder@uct.ac.za (NJM); jonathan.blackburn@uct.ac.za (JMB)

**Data Availability Statement:** The Human mucosal-luminal interface mass spectrometry proteomics data are available from the ProteomeXchange Consortium via the PRIDE partner repository with the dataset identifier PXD003528, and all results

## Abstract

### Background

Microbiome research is providing important new insights into the metabolic interactions of complex microbial ecosystems involved in fields as diverse as the pathogenesis of human diseases, agriculture and climate change. Poor correlations typically observed between RNA and protein expression datasets make it hard to accurately infer microbial protein synthesis from metagenomic data. Additionally, mass spectrometry-based metaproteomic analyses typically rely on focused search sequence databases based on prior knowledge for protein identification that may not represent all the proteins present in a set of samples. Metagenomic 16S rRNA sequencing only targets the bacterial component, while whole genome sequencing is at best an indirect measure of expressed proteomes. Here we describe a novel approach, **MetaNovo**, that combines existing open-source software tools to perform scalable *de novo* sequence tag matching with a novel algorithm for probabilistic optimization of the entire **UniProt** knowledgebase to create tailored sequence databases for target-decoy searches directly at the proteome level, enabling metaproteomic analyses without prior expectation of sample composition or metagenomic data generation and compatible with standard downstream analysis pipelines.

### Results

We compared **MetaNovo** to published results from the **MetaPro-IQ** pipeline on 8 human mucosal-luminal interface samples, with comparable numbers of peptide and protein identifications, many shared peptide sequences and a similar bacterial taxonomic distribution compared to that found using a matched metagenome sequence database—but simultaneously identified many more non-bacterial peptides than the previous approaches. **MetaNovo** was also benchmarked on samples of known microbial composition against matched

have been uploaded to the ProteomeXchange Consortium via the PRIDE partner repository with the dataset identifier PXD030708 (https://www.ebi.ac.uk/pride/archive/projects/PXD030708). The MetaNovo software is available from GitHub (https://github.com/uct-cbio/proteomics-pipelines/releases/tag/v1.7.0) and can be run as a standalone Singularity or Docker container available from the Docker Hub (https://hub.docker.com/r/thyscbio/metanovo). The companion code used to generate the analysis for this manuscript are available in a dedicated repository on GitHub as jupyter notebooks (https://github.com/Thys3Potgieter/Metanovo-Manuscript).

**Funding:** MGP would like to thank the National Research Foundation (NRF) of South Africa for an MSc grant (NRF BFG 93665). DLT was supported by the South African Tuberculosis Bioinformatics Initiative (SATBBI), a Strategic Health Innovation Partnership grant from the South African Medical Research Council and South African Department of Science and Technology. JMB thanks the NRF for a South African Research Chair grant. The funders had no role in study design, data collection and analysis, decision to publish, or preparation of the manuscript.

**Competing interests:** The authors declare that they have no competing interests.

metagenomic and whole genomic sequence database workflows, yielding many more MS/MS identifications for the expected taxa, with improved taxonomic representation, while also highlighting previously described genome sequencing quality concerns for one of the organisms, and identifying an experimental sample contaminant without prior expectation.

## Conclusions

By estimating taxonomic and peptide level information directly on microbiome samples from tandem mass spectrometry data, **MetaNovo** enables the simultaneous identification of peptides from all domains of life in metaproteome samples, bypassing the need for curated sequence databases to search. We show that the **MetaNovo** approach to mass spectrometry metaproteomics is more accurate than current gold standard approaches of tailored or matched genomic sequence database searches, can identify sample contaminants without prior expectation and yields insights into previously unidentified metaproteomic signals, building on the potential for complex mass spectrometry metaproteomic data to speak for itself.

## Author summary

**MetaNovo** is an open-source software pipeline that integrates existing tools with a custom algorithm to produce targeted protein sequence databases for mass spectrometry metaproteomic analysis as an intermediate filtering step prior to standard sequence database search approaches. **MetaNovo** uses open-source tools to match peptide mass spectrometry spectra to sequence database entries in a parallelised and scalable manner and can be installed in a cluster or run standalone on a Linux machine. The software is scalable to the number of input files and search sequence database size. As inputs the software requires raw mass spectrometry data in MGF format, and a **UniProt** FASTA sequence database to search. The pipeline is relevant to users analysing protein data from multiple organisms, where the exact species composition is unknown, such as microbiome or environmental samples, and provides an avenue for analysis when matched metagenomics data or accurate taxonomic characterisation is not available as it infers the organisms and proteins present directly from the raw data and the parent sequence database. The targeted sequence database produced can be used with standard downstream peptide identification software that relies on a targeted input sequence database to search the raw data against and allows greater sensitivity in peptide spectral matching in metaproteomic datasets.

This is a *PLOS Computational Biology* Software paper.

## Introduction

Characterising microbial ecosystems from clinical or environmental samples promises insights into the complex metabolic pathways involved in processes as diverse as carbon sequestration and climate change to the susceptibility and progression of human diseases [1]. Microbial communities play a wide-ranging role in the complex determinants of human well-being, with potential disease implications should the balance be disrupted [2].

Genome and transcriptome sequencing approaches to microbiome analysis allow for functional and taxonomic characterization of the genes and organisms involved in complex microbial communities, but it has been shown that measures of gene transcription do not generally correlate well with measured protein abundance [3]. On the other hand, metaproteomic and clinical proteomic approaches in principle allow researchers to obtain a snapshot of all the proteins present in a complex microbiome sample at a given time, providing a quantitative window into key functional components of complex pathways and organism interactions, including the direct measurement of dynamic changes in microbial protein composition, localisation and modification that may mediate host/pathogen interactions in the context of human health and disease.

Classically, the identification of peptides (and by inference, the parent proteins) from complex mass spectrometry datasets has been largely dependent on the availability of focussed, representative and relatively small sequence databases or spectral libraries against which to search tandem mass spectra—perhaps due in part to the historical emphasis in the field on the analysis of model organism proteomes to answer simple biological questions. However, the analysis of complex, multi-species samples–such as human microbiome samples where the total number of microbial genes may vastly exceed the number of human genes—is far less straightforward for a variety of reasons, including: the vast majority of specific organisms in any given microbiome are likely to not have been cultured, identified, or characterised in the laboratory, so appropriate reference genomes may not exist to underpin proteomic data analyses; and the fact that massive expansion of proteomic sequence databases used in target-decoy-based peptide-spectrum matching has been shown to lead to dramatically reduced identification rates due to higher rates of false negatives and false discovery misestimation problems [4].

Various methods have been developed to address the challenges of metaproteomic MS/MS identification. Previous methods for metaproteomic analysis have addressed these issues essentially by relying on prior expectation of the microbial composition of a given microbiome sample, for example using 16S sequencing data to produce a focussed metaproteome sequence database to search [5]. However, some limitations of this approach are that: (i) it assumes that the 16S data itself provides a complete description of all bacteria present in the sample, which is probably not true; (ii) it assumes that appropriate reference bacterial proteomes exist and are close enough in sequence to the clinical isolate proteomes for proteomic data analysis to be valid, which may also not be true; (iii) it automatically ignores all non-bacterial components of the microbiome; & (iv) the expressed proteome even within a single species is dynamic and not constant under all conditions, leading to the likely inclusion of spurious protein sequences even for correctly included organisms, thus inflating the search space and leading to decreased sensitivity. These approaches also imply additional analyses that depend on resources and samples that may not be available in all cases. Whole metagenome sequencing itself also suffers from the complexity of short reads leading to fragmented contigs and scaffolds which may interrupt protein-coding regions and hamper CDS predictions. Tang et al. [6] used *de Bruijn* graphs generated by metagenome assembly to produce protein sequence databases for metaproteomics, where identified peptides may span separate contigs in the graph, yielding many more peptide identifications than linear approaches to metagenome data.

Iterative "two-step" search strategies—where matches from a primary search against an initially very large sequence database at relaxed false discovery rates (FDR) are used to create a focused sequence database for subsequent more stringent target-decoy search—have been used, for example by the **MetaPro-IQ** pipeline, to yield a higher number of identifications [7,8], but there are some concerns about the statistical implication of re-searching data repeatedly using identical or closely related algorithms.

Further pitfalls of the use of targeted sequence databases for metaproteomics include the danger of mis- assignments when proteins truly present in the data are not included in the search sequence database. A case in point are spectra assigned to viral proteins identified and published in a study but later identified as *Apis mellifera* honey bee proteins by another group —that may have been differently assigned if the original authors included those proteins in the original search sequence database [9], underlining a significant potential problem if metapro-teomics studies ignore the presence of viruses, archaea, fungi and dietary components that are likely to be present in a sample.

Reanalysis of the same *Apis mellifera* samples by another group [10], found that only a small percentage of the viral peptides could be explained by homologous peptides, and argued that limiting the search space to only peptides of interest, reduces multiple hypothesis testing, and thus increases peptide identification sensitivity. However, they pointed out that leaving out sequences from the search sequence database could lead to false assignments of spectra of true proteins to biologically related sequences in the sequence database. In the case of metapro-teomic analysis representing multiple species with potentially many orthologous proteins, arti-ficially limiting the search space may therefore lead to false positive peptide identifications despite decreased false negative identifications. However, reducing the search space to peptides of interest would be an interesting avenue for analysing complex samples, but requires a repre-sentative sequence database that is available for filtering, and prior knowledge of sample com-position, and does not work for very small sequence database sizes, due to false discovery estimation problems.

The ideal sequence database for mass spectrometry-based peptide identification in meta-proteomic datasets would therefore be comprehensive, whilst excluding proteins that are gen-uinely absent, and would not be reliant on prior expectation of sample composition, nor would it require prior generation of DNA/RNA sequencing data to inform the construction of a search sequence database.

Taxonomic profiling of proteins identified during metaproteomic analysis is also compli-cated by high levels of shared tryptic peptides between homologous proteins of closely related organisms. To help solve this, the **UniPept** *pept2lca* algorithm [11] assigns the lowest common ancestor (LCA) for a given peptide assignment, whilst the recently published **ProteoClade** algorithm [12] allows users to apply species level annotation to identified peptides using cus-tomised search sequence databases.

*De novo* sequencing of peptides from mass spectrometry data has long been used for sequence database filtration, allowing for rapid searches of very large search spaces with sequence tags prior to peptide-spectral matching [13]. Tanner *et al*. published the **InsPecT** tool in 2005 [14]. These approaches work by reducing the sequence database size by orders of magnitude, while retaining the correct sequences with high probability. The sequence tag fil-tering approach is interesting, as in isolation full peptide sequencing of higher-energy colli-sional dissociation (HCD) data has been found to be only around 35% accurate [15]. Moreover, incomplete fragmentation of experimental spectra hampers accurate full-length sequencing using *de novo* algorithms [16]. Notwithstanding these challenges, partial sequenc-ing with sequence tag approaches with tools such as **DirecTag** [17] represents a plausible, robust and scalable approach for sequence database filtering, whilst recent advances in deep-learning-based approaches have significantly improved the accuracy of full sequence *de novo* sequencing approaches, with tools such as **pNovo3** yielding up to 89% improvement in preci-sion compared to other state-of-the-art tools, illustrating the current rapid progress in this field [18].

Here, we present a novel metaproteomic analysis pipeline—**MetaNovo**–improving on the conventional methodology for iterative sequence database search by mapping *de novo*

sequence tags to very large protein sequence databases using a high-performance and parallelized computing pipeline as the first search, thereby generating compact sequence databases for subsequent, orthogonal peptide-spectrum matching (PSM)-based target-decoy analysis and FDR-controlled protein identifications. We compare the results of **MetaNovo** against those published for the two-step **MetaPro-IQ** pipeline using a publicly available mass spectrometry dataset. We further validate the pipeline on samples of known microbial composition as ground truth. To our knowledge, this is the first published metaproteomic workflow for iterative two-step searching that uses *de novo* sequencing and probabilistic protein inference as a first step, followed by orthogonal conventional target-decoy search with FDR control in the second step. We show that replacing conventional PSM-based sequence database searching with *de novo* sequencing in the first step extends the applicability of iterative searches to extremely large sequence databases, which have previously only been applied to large, curated lists of proteins, thus decreasing the bias inherent in the analysis of complex metaproteomic data using these approaches.

The **MetaNovo** software has already been used to obtain new insights into vaginal [19,20] and foreskin [21] microbiome dynamics and has been made publicly available on **Galaxy** [22] (https://toolshed.g2.bx.psu.edu/repository?repository_id=f90d438ce91f95f1, https://usegalaxy. eu/root?tool_id=metanovo,). The tool is also available as a standalone package in **Anaconda** (https://anaconda.org/bioconda/metanovo).

## Design and implementation

### *MetaNovo* sequence database generation workflow and benchmarking

**UniProt sequence database statistics.**   The 2019_11 release [23] of **UniProt** was downloaded, and a combined FASTA sequence database created from **UniProt SwissProt**, **TREMBL**, and **SwissProt** alternative splice variant sequences (ca. 180 million sequences). See *S1 Table*.

### General software overview

The **MetaNovo** software is a configurable command-line Linux **Bash** pipeline that combines open-source software tools with custom **Python** libraries and scripts to provide targeted protein identification search libraries in **FASTA** format. The pipeline has three main components based on custom and existing open-source tools—generating *de novo* sequence tags using **DirecTag**, mapping the tags to a protein sequence database using **PeptideMapper**, and a novel algorithm for probabilistic protein ranking and filtering based on estimated species and protein abundance. The **MetaNovo** software is available from **GitHub** [24] and can be run as a standalone **Singularity** or **Docker** container available from the **Docker Hub** [25]. The Human mucosal-luminal interface mass spectrometry proteomics data are available from the **ProteomeXchange Consortium** via the **PRIDE** [26] partner repository with the dataset identifier **PXD003528**, and all results have been uploaded to the **ProteomeXchange** Consortium via the **PRIDE** partner repository with the dataset identifier **PXD030708**. *See Fig 1. The Typical **Meta-Novo** workflow.*

### Generation of *de novo* sequence tags

Raw MS/MS files need to be converted to *Mascot Generic Feature* (MGF) format prior to analysis. *De novo* sequence tags are generated using **DeNovoGUI** version 1.15.11 [27], with **Direc-Tag** [17] selected as the sequencing engine. General **MetaNovo** parameters selected for the analyses include fragment and precursor ion mass tolerance of 0.02 Da, with fixed

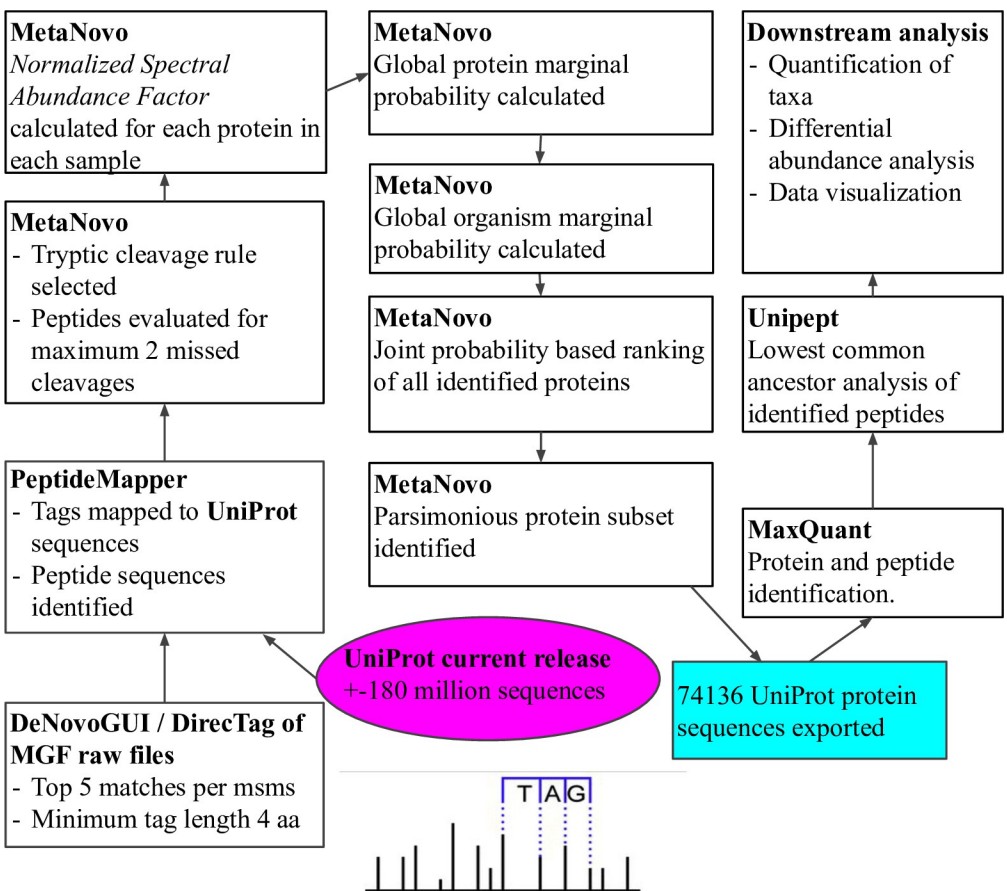

**Fig 1. Visualisation of the MetaNovo workflow used to analyse the mass spectrometry data of 8 human mucosal-luminal interface samples.** Raw mass-spectrometry data were analysed using the **MetaNovo** pipeline in MGF format, using *de novo* sequence tags to create a targeted FASTA file for target-decoy search.

modification '*Carbamidomethylation of C*' and variable modifications '*Oxidation of M*' and '*Acetylation of protein N-term*'. For **DirecTag**, a tag sequence length of 4 amino acids was required, and the top 5 sequence tags per spectrum were selected (taking into account alternative possible charge states). These settings were chosen as preliminary runs with a sequence tag length of 3 proved impractical with very large sequence databases, requiring more than 5 days with 24 threads on a high-performance cluster to search 8 MGF files against **UniProt**. Similarly, increasing sequence tag options per spectrum from 5 to 30 considerably increased processing time without a corresponding gain in sensitivity, possibly due to the inclusion of low-scoring matches. The output of **DirecTag** is parsed with a custom **Python** script and all sequence tags for each MS/MS across replicates are stored in an **SQLite** database [28]. The distinct set of sequence tags (N-terminus mass gap, amino acid sequence, and C-terminus mass gap) are obtained using an **SQL** query, combining identical tags across multiple MS/MS into a single non-redundant list.

## Mapping sequence tags to a FASTA sequence database

**PeptideMapper** [29] included in **compomics-utilities** version 4.11.19 [30] is used to search the sequence tag set against the protein sequence database. The same mass error tolerance and post-translational modification settings are used as for **DeNovoGUI** and specified in the config

file. The open-source **GNU parallel** tool [31] is used to search the set of sequence tags against the FASTA sequence database in configurable chunk sizes, using a configurable number of threads per node in parallel.

### Enzymatic cleavage rule evaluation

A custom **Python** script is used to evaluate the cleavage rule of the peptide sequences of identified sequence tags. Only peptides passing the selected cleavage rule are selected for downstream analysis ('Trypsin, no P rule' was selected for the benchmarking analysis—cleavage after all *Lys* or *Arg*) with up to 2 missed cleavages allowed. The corresponding sequence tags for the validated peptide sequences are queried against the **SQLite** database to obtain a mapping of MS/MS ids to protein ids, allowing for the estimation of protein abundance based on mapped MS/MS in a similar manner to spectral counting in a target-decoy search.

### Normalised spectral abundance factor calculation

Protein abundance is estimated using the *Normalised Spectral Abundance Factor* (NSAF) approach developed for shotgun proteomics [32]. The set of MS/MS IDs in each sample per protein is obtained, and the number of mapped MS/MS ids is divided by protein length to obtain a *Spectral Abundance Factor* (SAF). The SAF values for each sample are divided by the sum of the SAF values in that sample, to obtain an NSAF for each protein in each sample/replicate.

### Probabilistic ranking of sequence database proteins

The **MetaNovo** algorithm uses the concept of unconditional or marginal probability, which is a probabilistic value generated for each protein in the **UniProt** sequence database corresponding to the probability that a given peptide fragment sampled at random from the raw data will belong to that protein given the available data. To calculate the marginal probabilities, protein NSAF values are summed across replicates, and divided by the sum of all NSAF values across all replicates, with a maximum value of 1 for each protein. This value is equivalent to the proportion that a given protein makes up of all replicates by summed NSAF value. SQLite database proteins are ranked by this marginal probability and filtered such that a minimal set of ranked proteins is obtained where each protein in the list contains at least one unique MS/MS scan ID relative to the set of proteins above its position in the list—and excluding all proteins that do not contain any uniquely identified spectra relative to the set of proteins above their position in the ranked list. Using this methodology, sets of protein isoforms that share an exact set of mapped MS/MS ids will be represented by the shortest sequence in the set due to having a higher NSAF value, based on the principle of Occam's razor where longer sequences are more likely to have sequence tag matches by pure chance. The organism names are obtained from the **UniProt** FASTA headers, and summed NSAF values in the filtered set are aggregated by organism and divided by the total to obtain an organism-level marginal probability. Following this, the proteins in the unfiltered set are annotated with their respective calculated organism marginal probability. The unfiltered set of proteins is re-ranked based on the joint probability of the organism and protein probability values, based on the simplistic assumption of conditional independence (Naive Bayes Assumption) which allows low abundance proteins with a high estimated relative abundance at the organism level to be favoured compared to proteins where both the protein and organism abundance are estimated to be low. Benefits of the Naive Bayes assumption of conditional independence between predictors are ease of mathematical implementation and being highly scalable to the number of predictors and data points, without sacrificing model accuracy, and allowing model training and classification

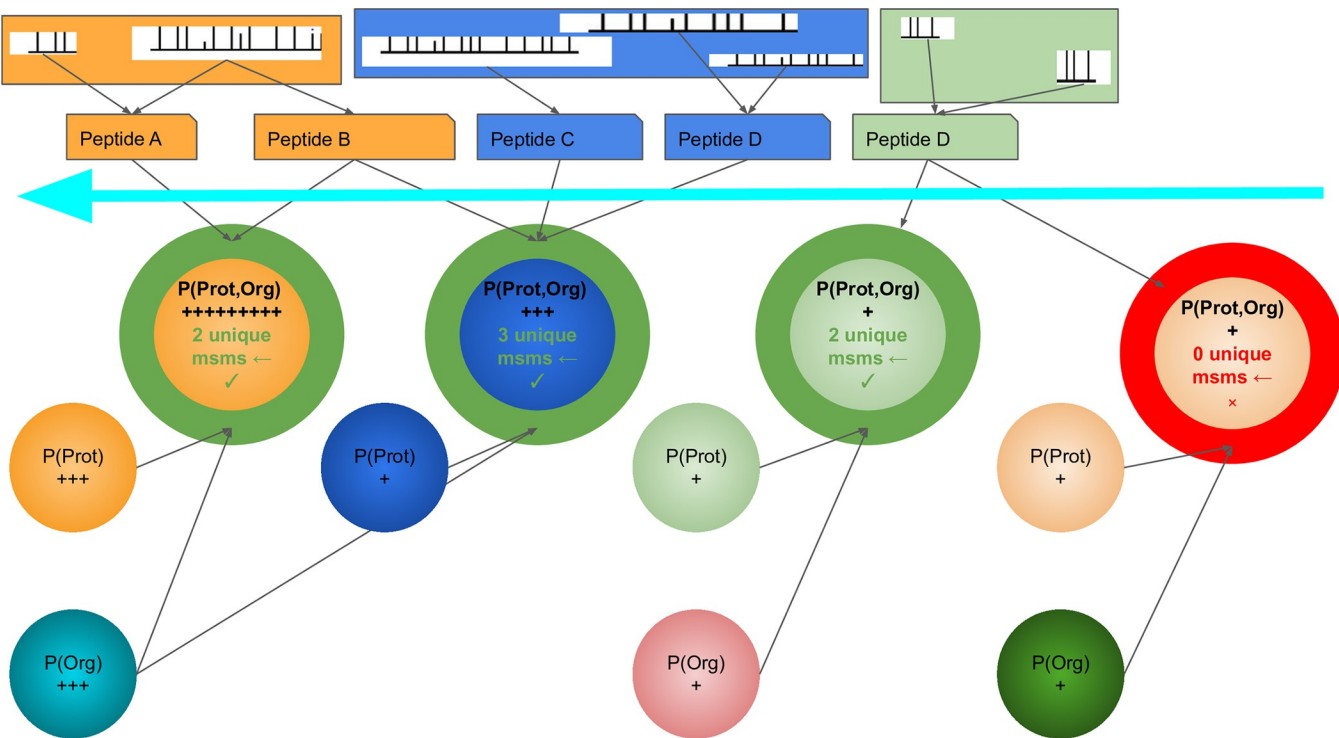

**Fig 2. A Graphical representation of the MetaNovo algorithm applied for sequence database filtration.** *Normalized spectral abundance factor* calculations include non-unique spectra. The magnitude of probabilities are represented by +'s. Proteins are ranked by the joint probability of organism and protein probabilities, represented by the arrow, in order of increasing probability. The number of unique spectra for each protein is determined based on its position in the ranked list, and only include spectra that do not appear in the set of proteins in the list above (but may include spectra that appear below), such as the spectra for **Peptide B** that are counted towards the first protein in the list, but not the second. Tie breaks for adjacent and nearly identical isoforms that share the same set of spectra, will be based on the shortest (most probable) sequence having a higher NSAF (and thus a higher protein probability) or a higher organism probability. Proteins in green will be selected for inclusion in the filtered sequence database, and proteins in red will be excluded (having no unique spectra). The colors shared by proteins, peptides and spectra above, illustrate the assignment of unique spectra and peptides, to the most probable protein in the ranked list.

with a single pass over the data. The protein joint probability calculation used to rank protein ids in the SQLite database is as per the equation below:

$$P(Organism, Protein) = P(Organism) * P(Protein)$$

Following the second ranking step, the database is re-filtered to obtain a minimal subset of proteins that can explain all mapped *de novo* sequence tags, based on estimated relative protein and organism abundance. *See Fig 2*.

## Mucosal-luminal interface samples

**Mucosal-luminal interface (MLI) sample metagenomic and proteomics data.** Metagenome and proteomics data of 8 MLI samples from adolescent volunteers obtained during colonoscopy were downloaded from **PRIDE** with identifier **PXD003528** and through author correspondence. The sample processing, mass spectrometry and metagenomics sequence database creation methods have already been described [8].

*MetaNovo* **sequence database generation.** The December 2019 release [23] of **UniProt** (containing ca. 180 million protein entries) was used to create a sequence database containing 74136 entries. The **MetaNovo** settings described above were used. The experiment was run on a high performance computing cluster, with 120 GB available RAM, and 8 allocated Intel Xeon

Processor (Skylake, IBRS) cpus, and total running time was 3d03h38m to process the 8 MGF files.

## Sequence database search using MaxQuant

**MaxQuant** version 1.5.2.8 was used to search the **MetaNovo sequence** database, with the same settings as for the **MetaPro-IQ** publication. *Acetyl (Protein N-term)* and *Oxidation (M)* were selected as variable modifications, and *Carbamidomethyl (C)* was selected as a fixed modification. Specific enzyme mode with *Trypsin/P* was selected with up to 2 missed cleavages allowed, and a PSM and protein FDR of 0.01 was required.

Nine-organism microbial mixture samples.

## 9MM sample and validation sequence databases

Raw proteomic data from two samples of a known nine-organism microbial mixture (9MM) from a single biological replicate were downloaded from **PeptideAtlas** (identifier **PASS00194**) [33]. *See S2 Table*. Detailed proteomic methods were described in the original publication [33]. In brief, each of 9 organisms was cultured separately on the appropriate growing media and divided into aliquots of approximately $10^9$ CFU each. The 9MM sample was created by combining an aliquot of each organism pellet and processed separately by filter-aided sample preparation (FASP) and protein precipitation followed by in-solution digestion (PPID). Two sequence databases were selected from the original publication for comparison. One, the top-performing sequence database from the previous publication was created by single genome assembly followed by gene prediction and protein annotation using **TrEMBL**, with 27164 non-redundant entries (**SGA-PA**). Secondly, to illustrate a typical metaproteomics workflow, the sequence database created by NGS of the two 9MM extracts (metagenome sequencing) followed by gene prediction and protein annotation using **TrEMBL** was selected, with 13270 non-redundant entries (**Meta-PA**).

## *MetaNovo* sequence database generation

The December 2019 release [23] of **UniProt** (containing ca. 180 million protein entries) was used to create a sequence database containing 13195 entries. The **MetaNovo** settings described above were used. The experiment was run on a single node of a high-performance computing cluster (https://www.ilifu.ac.za/), with 120 GB available RAM, and 8 allocated Intel Xeon Processor (Skylake, IBRS) processors. The total running time was 0d09h11m to process the 2 MGF files.

## Sequence database search using MaxQuant

**MaxQuant** version 1.5.2.8 was used to search the **MetaNovo, SGA-PA** and **Meta-PA** databases using the same search parameters. *Acetyl (Protein N-term)* and *Oxidation (M)* were selected as variable modifications, and *Carbamidomethyl (C)* was selected as a fixed modification. Specific enzyme mode with *Trypsin/P* was selected with up to 2 missed cleavages allowed, and a PSM and protein FDR of 0.01 was required. The FASP and PPID samples were treated as separate experiments.

## Bioinformatic analysis

**Posterior error probability (PEP) score analysis.**    PEP scores are a standard metric to identify and rank the quality of PSMs and are used during the FDR calculation of target decoy searches, as false-positive PSMs tend to rank similarly to reverse or decoy hits, allowing a

convenient method to estimate the FDR at a given position in the PEP score ranking [34]. PEP scores have been applied in the field of proteogenomics to evaluate the accuracy of novel peptide identifications, by comparing their distribution to that of reverse hits, allowing for the assessment of the level of completeness of protein annotation in a given strain, as novel peptide identifications in completely annotated genomes are likely false positive identifications [35]. The PEP scores of PSMs obtained using **MaxQuant** were obtained from the *peptides.txt* output files of the different sequence database runs of the MLI samples—using **MetaNovo**, **MetaPro-IQ** using the integrated gene catalog (IGC) of human gut microbial genes [36], hereafter referred to as **MetaPro-IQ/IGC**, and a matched metagenome sequence database hereafter referred to as **MetaPro-IQ/Metagenome** (the published **MaxQuant** results using the **Meta-Pro-IQ/IGC** and **MetaPro-IQ/Metagenome** sequence databases were obtained from PRIDE). PSMs for each run were grouped into "exclusive" peptide sequences only identified in that run, and PSMs of "shared" peptides that were also identified in the other two runs, as well as the reverse hits for each group. Significant differences between the groups were tested for using the *Kruskal-Wallis* non-parametric analysis of variance test followed by *Dunn's* post hoc test using a custom python script. PEP score distributions were visualised using the python *matplotlib* library.

## Taxonomic analysis

Peptide sequences were assigned to the lowest common ancestor (taxonomic level) using the **UniPept** *pept2lca* tool, to allow for a comparison between the results of the different approaches and to the ground truth samples of known taxonomic composition (9MM) [33]. Peptide and spectral counts were aggregated at selected phylum levels reported in the *pept2lca* output, with custom visualisations done using the Python *matplotlib* module. We followed a previously described approach to filtering spurious taxa based on **UniPept** results by only including taxa that represent more than 0.5% of taxonomically characterised peptides [33], applied separately at each taxonomic level.

## BLAST analysis of selected peptides and proteins from 9MM samples

Peptides identified in the 9MM analysis that were assigned using the **UniPept** *pept2lca* tool to organisms not expected from the known mixture were selectively subjected to **NCBI BLAST** analysis. Leading proteins for the same peptides were extracted from the *proteinGroups.txt* file and the protein sequences submitted for **NCBI BLAST** [37]. Proteotypic peptides were defined as only matching to a single species with 100% identity in the **NCBI** nr sequence database. Species-specific protein sequences were defined as those sequences that only matched to a single species above a threshold of 90% identity in the **NCBI** nr sequence database.

## 9MM Taxonomic representation error scores calculation

Spectral counts obtained from the **MaxQuant** output files were aggregated to *family*, *genus* and *species* levels for the expected taxa. For each expected taxon, the percentage relative to the total characterised MS/MS (including mis assignments) was calculated for the **MetaNovo**, **SGA-PA**, and **Meta-PA** workflows respectively. As approximately equal proportions of the 9 organisms were analysed based on CFU counts, ground truth percentages were assigned to the 3 taxonomic levels accordingly. Mean Squared Error (MSE) scores were calculated for relative taxon abundance for each of the three runs relative to the ground truth percentages and compared. MSE values closer to 0 indicate lower error rates, and therefore a more accurate representation of relative taxon abundance in the samples.

## Results

### MetaNovo identifies non-bacterial taxa, whilst providing comparable bacterial composition data to a matched metagenome approach in human Mucosal-luminal interface samples

**Protein and peptide identifications.** The **MetaNovo** pipeline was run using the entire, unfiltered **UniProt** sequence database (December 2019 release; ca. 180 million entries), resulting in the identification of 69878 target peptides and 15731 protein groups, with 36.37% of total MS/MS identified. These results are comparable but slightly better with >2% more spectra assigned than those of previous approaches, with 34 and 33% MS/MS identification rates reported by the **MetaPro-IQ** authors using a matched metagenome or an integrated gene catalog (IGC), respectively. 48705 peptides were identified in common by all three runs, while 4873, 6525 and 14049 were identified exclusively by the **MetaPro-IQ/IGC**, **MetaPro-IQ/Metagenome**, and **MetaNovo** sequence databases, respectively. *See Fig 3B*.

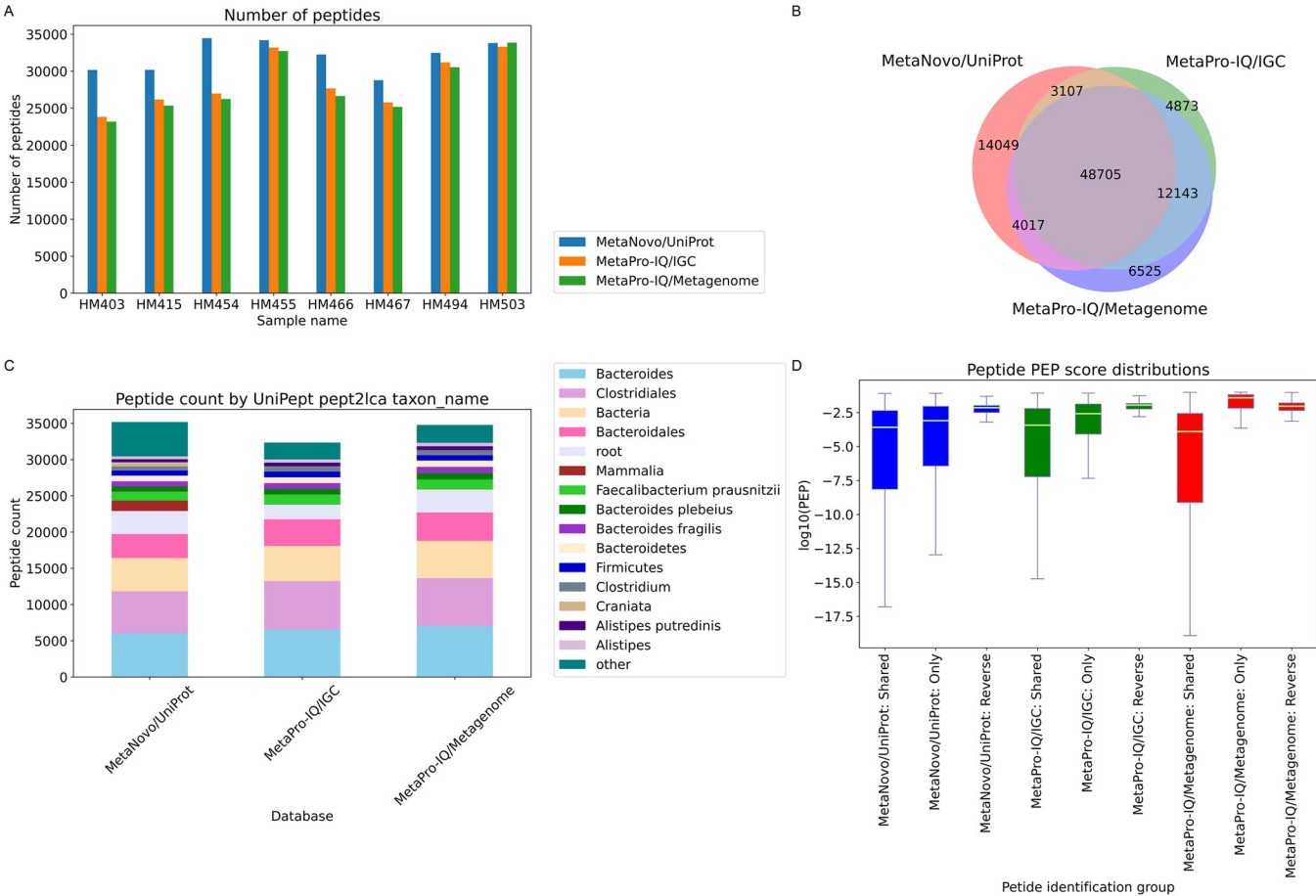

**Fig 3. MLI dataset results. A.** Bar chart of peptide identifications. The identification rates of **MetaNovo** are comparable to the previously published results of **MetaPro-IQ** using matched metagenome and *integrated gene catalog sequence* databases. **B.** Venn diagram showing large overlap in identified sequences using different approaches, with the highest number of sequences identified using **MetaNovo. C.** Peptide counts by *UniPept lowest common ancestor* showed similar taxonomic distributions obtained from different approaches. **D.** Peptides uniquely identified by **MaxQuant** using the **MetaNovo sequence** database had a significantly different distribution compared to reverse hits (p-value 6.33e-26). The boxes extend from the lower to the upper quartile, and the whiskers represent 1.5 times the interquartile range (IQR) below and above the first and third quartiles, respectively.

## PEP score analysis

Kruskal-Wallis analysis of variance howed a statistiucally significant difference between exlusive, shared and reverse hit PEP scores across the three runs (statistic = 10883.94, pvalue = 0.0). Visual inspection of PEP score density plots shows a similar distribution of peptide PEP scores for peptides only identified in a specific run, and peptides that were identified in all runs, with the exception of peptides only identified using the **MetaPro-IQ/Metagenome sequence** database, that showed a very broad distribution overlapping with the reverse hits (*See S1 Fig*). Analysis of shared and exclusive peptide identifications in each group revealed significantly different distributions between 14049 peptides only identified using **MetaNovo** and reverse hits for the same run (p-value 6.33e-26), indicating true positive identifications in this set not found using the other approaches. The difference between peptides exclusively identified with the **MetaPro-IQ/IGC** sequence database and reverse hits was also significant (p-value 3.15e-20). The 6525 peptides only identified using the matched metagenome had a higher median PEP score (0.038306) compared to reverse hits (0.009337) although the difference between the two groups was not statistically significant(p-value 4.60e-01), suggesting a higher rate of false-positive hits in this group than the other two approaches. *See Fig 3D and S3 and S4 Tables*.

## Taxonomic comparisons

Peptide counts by lowest common ancestor were compared between different analysis runs using the output of UniPept *pept2lca (See S5 Table)*. After applying stringency criteria based on requiring at least 0.5% of total classified peptides per included taxon, the highest number of characterised peptides by UniPept pept2lca *taxon_name (the most specific possible taxonomic characterization for a given peptide across all taxonomic levels)* was obtained using **MetaNovo/ UniProt** (35169), followed by **MetaPro-IQ/Metagenome** (34768) and **MetaPro-IQ/IGC** (32325). A similar taxonomic distribution was obtained between runs (*See Fig 3C)*. Of the characterised peptides passing the stringency criteria, 4700 peptides belonged to *taxon_name* taxa that were identified exclusively by **MetaNovo**. These taxa are consistent with expected host peptides that were not identified by the other runs (no *Homo sapiens* peptides were identified by the other approaches). *See S6 Table*. Conversely, all taxa identified by the other analysis methods were also identified by **MetaNovo**. *See S7 Table*. When selecting only Bacteria "*super-kingdom_name*" peptides, lower numbers of peptides were identified by **MetaNovo** than for the other runs, with 27289, 30311 and 31591 annotated peptides for **MetaNovo**, **MetaPro-IQ/ IGC**, and **MetaPro-IQ/Metagenome**, respectively. A similar taxonomic distribution was visualised for all three runs in this group. *See S2 Fig*. When aggregating peptide counts at the "*kingdom_name*" level, peptides from *Metazoa*, *Viridiplantae* and *Fungi* kingdoms were represented by all runs, but with much higher numbers represented by **MetaNovo**. These taxa are all known to play key roles in the human gut microbiome, as living or dietary components. *See S8 Table*. Similarly, aggregated to *Phylum* level, **MetaNovo** identified the same phyla as identified in either of the other runs, but with much higher numbers of *Chordata* peptides. Interestingly, both the **MetaNovo/UniProt** and **MetaPro-IQ/IGC sequence** databases identified *Actinobacteria*—but these were not identified by the **MetaPro-IQ/Metagenome sequence** database. *See S9 Table*.

## Analysis of a known microbial mixture using the MetaNovo software yields increased sensitivity compared to matched genomics sequence databases

**Protein and peptide identifications.** The **MetaNovo** pipeline was run using the entire, unfiltered **UniProt sequence** database (December 2019 release; ca. 180 million entries),

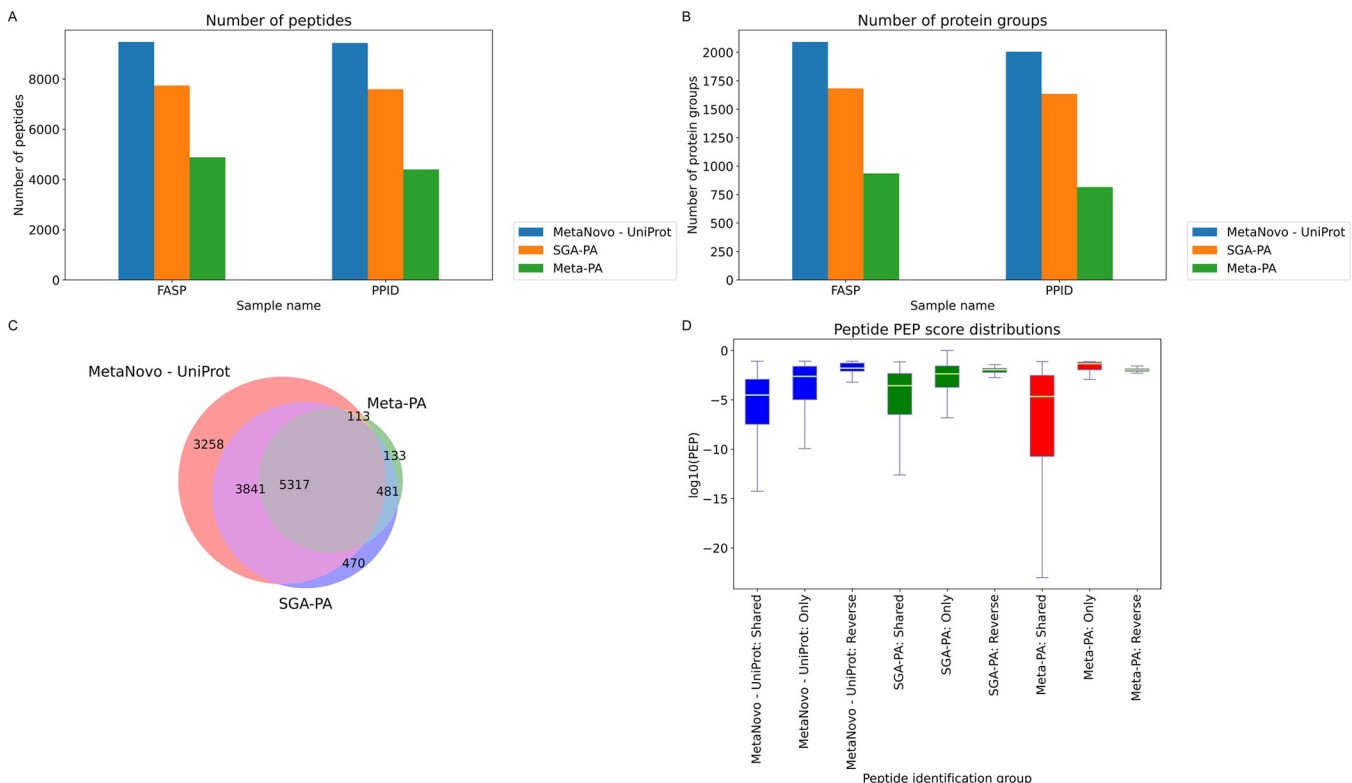

**Fig 4. 9MM dataset identification results. A.** Number of peptides identified in each run. **B.** Number of protein groups identified in each run. **C.** Peptide identification overlap between the different approaches. **D.** Peptide PEP score distribution box plot for shared, exclusive and reverse hit peptides for each run.

resulting in the identification of 12529 target peptides and 2534 target protein groups, with 44.17% of total MS/MS identified. These results are significantly better with 9.14% more spectra assigned than the best performing genomic sequence database reported by the original authors when run with the same **MaxQuant** settings (single predicted and annotated genomes assembly DB (SGA-PA)) that yielded 35.03% identified MSMS, 10109 target peptides and 2099 target protein groups. The results were almost double those of the metagenome (Meta-PA) sequence database that yielded 23.03% identified MSMS, 6044 target peptides and 1087 target proteins (*See Fig 4A*), despite a similar size of the Meta-PA sequence database to the MetaNovo sequence database. Kruskal-Wallis analysis of variance indicated a significant difference in the PEP score distribution of the exclusive, shared and reverse peptide matches of the three runs (statistic = 1525.73, pvalue = 0.0). The median PEP scores of the 3258 and 470 peptides exclusively identified using the MetaNovo and SGA-PA sequence databases were lower and the distributions significantly different to that of reverse hits for the same runs (p-value 1.05e-05 and 3.07e-02, respectively). The median PEP score of the 133 peptides exclusively identified by the Meta-PA sequence database (0.046675) was worse than that of the 11 reverse hit peptides of the same run (0.010709), but the difference was not statistically significant (p-value 8.11e-01). *See S10 and S11 Tables.*

## Taxon identification rates

Percentages of misassigned peptides were calculated for all three runs. **MetaNovo** originally yielded a very high percentage of misassigned spectra at species level based on **UniPept** *pept2lca* analysis. Further analysis of the characterised peptides indicated that most belonged

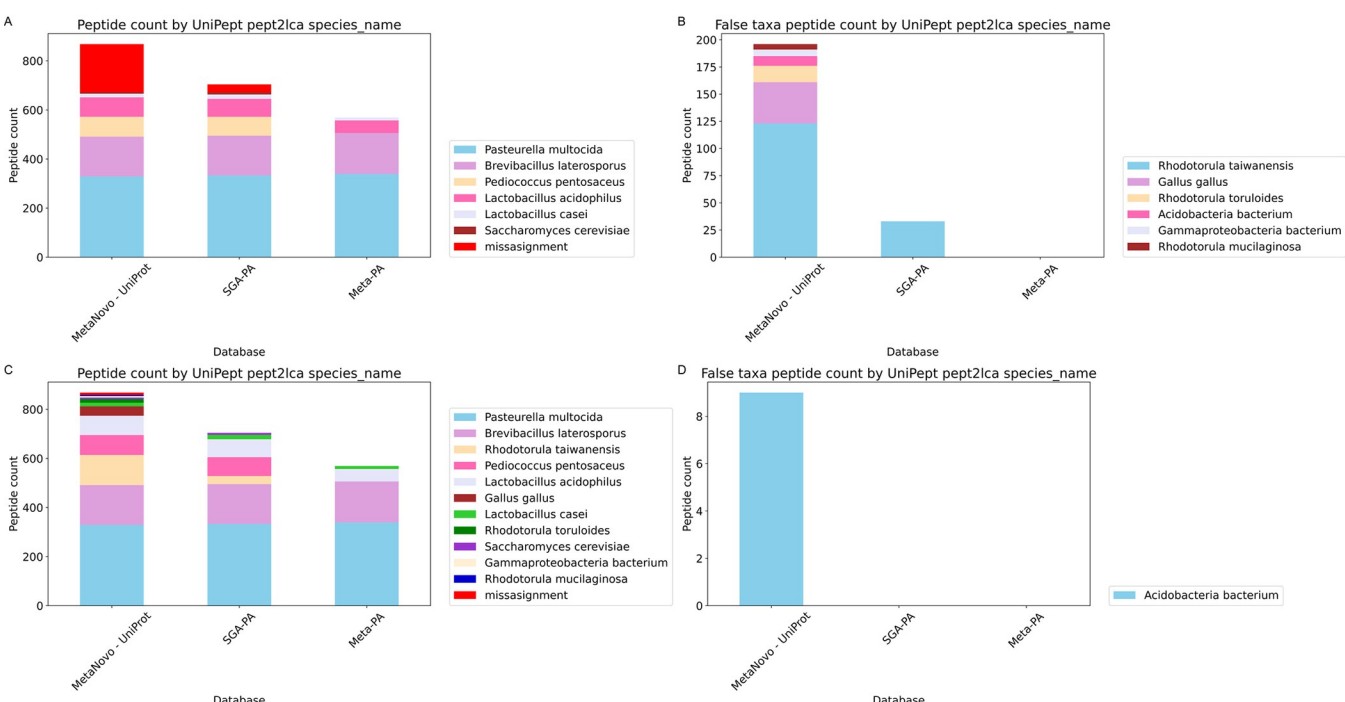

**Fig 5. Percentages of misassigned peptides for all three 9MM runs. A. MetaNovo** originally yielded a very high percentage of misassigned peptides at species level **UniPept** *pept2lca* analysis. **B.** Taxonomic breakdown of misassigned peptides **C.** Re-analysis after inclusion of plausible taxa yielded a species-level misassignment rate of only 1.04%, with 0% error for all approaches at genus and family level using the 0.5% taxon-specific peptide stringency cutoff. **D.** 9 *Acidobacteria bacterium* peptides making up the final misassignment percentage of **MetaNovo**.

to other *Rhodotorula* species, which corresponds to the authors' original report of many non-canonical peptides identified in their samples using a 6-frame translated sequence database, which could lead to **UniPept** assignment to closely related organisms. The identification of 38 *Gallus gallus* peptides was narrowed down to components of egg-yolk using **BLASTP** analysis (See *Unexpected taxa BLAST analysis)*. Notably, subsequent personal correspondence with those authors confirmed that egg yolk was present in one of the growth media employed in the experiment. The *Pasteurella multocida* strain analysed in this study was a field isolate (author correspondence), and being a member of Gammaproteobacteria, natural sequence diversity in this strain might be compatible with a classification of *Gammaproteobacteria bacterium*. After allowing for other *Rhodotorula spp.*, *Gallus gallus* and *Gammaproteobacteria bacterium* peptides, a species-level misassignment rate of only 1.04% was attained, with 0% error for all sequence databases at genus and family level using the 0.5% taxon-specific peptide stringency cutoff. The 9 *Acidobacteria bacterium* peptides making up the final misassignment percentage of **MetaNovo** could be explained by the higher total number of identified peptides leading to more false hits at the same FDR, actual contaminants, or **UniPept** misassignments of natural variants in the data. *See Fig 5 and S12 Table*.

## Taxon assignment accuracy benchmarks based on mean-squared error show the highest accuracy for species-level assignments using MetaNovo

Mean Squared Error (MSE) scores were calculated for the relative proportion of MS/MS of each taxon to the total of each run compared to the expected proportion by CFU counting as a percentage. Scores closer to 0 indicate higher accuracy. **MetaNovo** had a poorer score than **SGA-PA** for genus level assignments, but the highest accuracy of the three runs for both family

**Table 1. MetaNovo yields the highest accuracy for species-level annotations compared to matched genomic databases.** Mean Squared Error (MSE) scores for the relative proportion of MSMS of each taxon to the total of each run compared to the expected proportion by CFU counting as a percentage. Scores closer to 0 indicate higher accuracy.

|  | Meta-PA | SGA-PA | MetaNovo—UniProt |
|---:|---|---|---|
| **Family** | 398.89 | 143.58 | 106.08 |
| **Genus** | 261.64 | 78.44 | 87.32 |
| **Species** | 333.17 | 207.77 | 151.05 |
| **Species (relabelled)** | 333.17 | 198.79 | 138.03 |

and species level assignments. **MetaNovo** outperformed the matched metagenomic sequence database (**Meta-PA)** at all three taxonomic levels. After relabelling all *Rhodotorula* peptides as *Rhodotorula spp.* for the comparison, a further improvement in MSE scores was obtained, corresponding to the author's report of identifying many non-canonical peptides using the translated 6 frame sequence database for the same organism; some of these assignments may be homologous and identical to canonical peptides in other species in the same genus, and therefore were assigned to those organisms by **UniPept**. *See Table 1.*

## Unexpected taxa BLAST analysis

Of the taxonomically assigned peptides across the three runs, 38 *Gallus gallus* proteins were the largest group of false assignments by the **MetaNovo** pipeline compared to the list of expected taxa. This group was chosen for protein **BLAST** analysis. For each peptide, BLAST results were filtered to only include exact matches (100% identity). For each of these records, the organism names were identified, and 20 peptides that only matched to *Gallus gallus* were obtained as proteotypic peptides for this species, providing strong evidence at sequence level for this species in the data. See *S1 Data and S13 Table*. A FASTA file of eight protein sequences corresponding to the FASTA headers of the protein groups of the *Gallus gallus* peptides was submitted for protein **BLAST**. For each query in the FASTA file, the set of organisms for all matches with more than 90% sequence identity was investigated—and 5 sequences matched only to *Gallus gallus*. These proteins include *Apovitellenin-1* (P02659), *Vitellogenin-1* (P87498, A0A1D5NUW2), *Vitellogenin-2* (P02845) and *Vitellogenin-3* (A0A3Q2U347). These are all known components of egg yolk. See *S2 Data and S14 Table*.

## Discussion

On re-analysis of a set of Human Mucosal-luminal interface samples, comparative identification rates were achieved, with a >2% increase in the number of MS/MS at the same FDR—but now simultaneously identifying multiple human origin peptides not assessed by the previous approaches—while not leaving out taxa identified by the other runs. The large number of exclusive peptides only identified by *MetaNovo*, can be explained by the previous approaches not including host sequences in the search sequence database, with potential misassignments of host peptides to microbial search sequence database entries for those runs—a known risk for search sequence databases that are not comprehensive (*c.f.* the *Apis mellifera* reanalysis example discussed previously) [9]. Any such spurious PSMs are likely to have a higher PEP score distribution, overlapping with those from the reverse hit sequence database (which are all by definition spurious sequences)—thus leading to a decreased rate of target peptide identifications at the given FDR cutoff. This explains the lower MS/MS % identification rate of the previous approaches. The improved error distribution of exclusive peptides identified using **MetaNovo** supports this view and contrasts with the error distribution of peptides exclusively identified by **MetaPro-IQ** with the metagenomics sequence database, the latter having a higher

median PEP score than and with a distribution not significantly different from the distribution of reverse hits in the same experiment. A similar situation was found for the 9MM samples, where the median PEP score obtained from the Meta-PA only group (0.046675) was higher than the median peptide PEP score for reverse hits (0.010709), but the p-value for the difference in the distributions was not statistically significant. It is reassuring that in no comparison where the median PEP score was higher than for reverse hits, was the difference in the distribution statistically significant. It is interesting, in each case, that it was the sequence database that yielded the poorest MS/MS identification rate, that yielded the poorest PEP scores for peptides exclusively identified using that sequence database (not statistically significantly different from reverse hits) for both the MetaPro-IQ/Metagenome, as well as the Meta-PA 9MM sequence database runs. We suggest that due to poor sequence database representativity, PSMs to incorrect sequences in the sequence database occur due to the non-inclusion of the correct sequences, with peptides exclusively identified by the incomplete sequence databases having much poorer peptide PEP scores than the same groups obtained from the other, more complete sequence databases, and peptides only identified by the most representative sequence databases (by MS/MS percentage assignment) having the best median PEP scores in this category. As **MetaNovo** did not identify any exclusive taxa other than host compared to the other MLI runs, but with higher numbers of viridiplantae and fungi taxa, it points to improved characterization of non-bacterial components while maintaining an accurate characterization of the bacterial component without over-estimating taxonomic diversity. These other components of the gut microbiome may yield key insights into complex inter-specific interactions that may not be evident with approaches that target only limited taxonomies. Lower numbers of Bacterial peptide identifications by **MetaNovo** compared to the other two approaches may be due to the fact that both previous sequence databases are more comprehensive at the Bacterial level, while the **MetaNovo** algorithm aims to be globally representative, and increase the percentage of assigned spectra overall. However, it may also reflect the absence of host proteins from the search space, potentially leading to mis-assignments. Future updates to the **MetaNovo** algorithm will aim to increase the comprehensiveness of the generated sequence database, including known, clinically relevant host polymorphisms and mutations, so that the output can compete with taxonomically focused sequence databases and further increase the global sensitivity of assignments.

Taxonomic assignment of peptides from the 9MM data to *Rhodotorula* species other than the expected strain by **UniPept**, for both **MetaNovo** (143 peptides) and **SGA-PA** (33 peptides), indicate the limitations of the **UniPept** approach that is based on the transfer of reference annotations to the natural sequence diversity of field isolates. The higher number of *Rhodotorula* peptides identified by **MetaNovo** are consistent with the lower sequence coverage reported for the *R. glutinis* strain (x12) used for the **SGA-PA sequence** database. It is notable that the **MetaNovo** approach was able to yield improved performance over full genome sequencing by leveraging sequence information of related strains without prior expectation.

The almost double MS/MS assignment rate of MetaNovo compared to the Meta-PA sequence database (based on NGS of the two 9MM extracts), despite both sequence databases being almost identical in size, is most likely due to the lack of yeast proteins in the Meta-PA sequence database, excluding two of the 9 species present in the samples from the sequence database search space. This is an example of the utility of an approach that doesnt rely on matched metagenomics data to construct the search sequence database, as these may not be available or of high enough quality.

The unexpected but confident identification here of proteotypic peptides for *Gallus gallus*, as well as the identification of 5 proteotypic protein groups all related to egg yolk, in the 9MM samples prompted us to conjecture that egg yolk might have been a specific component of the

media used for *in vitro* culture of one or more of the 9 microbial species; this was subsequently confirmed by the authors of the original 9MM study. Thus, the identification of putative contaminants that may have otherwise been missed during sequence database creation confirms the power of our approach, including in cases where rare and unknown components of a biological sample would otherwise be missed.

Despite higher overall detection sensitivity, slightly lower numbers of Bacteria "superkingdom_name" peptides dentified by **MetaNovo** than for the other MLI runs, appears to support the hypothesis of Noble [10] that reducing multiple hypotheis testing can increase the sensitivity for peptides of interest, by the greater performance of the Bacteria focused sequence databases in bacterial identifications. The greater overall sensitivity of the **MetaNovo** sequence database, could similarly be due to the inclusion of only the most probable proteins within the included species, thus reducing multiple hypothesis testing globally. It could be an interesting avenue for future work, to provide the ability to limit the search sequence database export by the **MetaNovo** algorithm to only taxa of interest, to further reduce multiple hypothesis testing, both during sequence database filtering from **UniProt**, and peptide identification using the filtered sequence database.

Higher rates of MS/MS assignments yielded by the **MetaNovo** approach compared to the matched genomic approaches (both **MetaPro-IQ/Metagenome** and **SGA-PA**) indicate the utility of the **MetaNovo** algorithm in leveraging publicly available taxonomic and sequence data without sacrificing search sensitivity. Notably, the original, entire unfiltered **UniProt** sequence database seems likely to be more representative of expressed proteomes and natural sequence diversity in clinical samples than sequence databases compiled from reference bacterial proteomes based on 16S rRNA data and which may not directly reflect the actual sequences present at the proteome level. Further, as gene transcription does not correlate well with protein translation, sequence database generation approaches that rely only on matched metagenomic data may lead to an unnecessarily large search space and therefore decreased sequence database search sensitivity. Indeed, **MetaNovo** does not rely on any prior genomic data on specific samples as a pre-requisite for generation of sequence databases to search the mass spectrometry data against. Instead, **MetaNovo** relies on the sum total of all known, curated protein sequences to provide the search space, which provides additional advantages in circumstances where genomic data is not available or where simultaneous analysis of host, viral, fungal, dietary and bacterial proteomes is needed. Importantly, the production by **MetaNovo** of a more representative but still focused sequence database reduces the potential for mis-assignment of MS/MS due to sequence database incompleteness, leading to an increase in sequence database search sensitivity and accuracy.

It can be argued that the use here of the **UniProt** (**SwissProt** plus **trEMBL**) sequence database still provides a degree of bias in the search space compared to larger sequence databases such as the **NCBI** nr protein sequence database that contain many more proteins and species from clinical and environmental sources. Moreover, the current reliance of **MetaNovo** on annotated sequences may miss novel sequence polymorphisms present in clinical isolates. Further work to tailor the **MetaNovo** algorithm to work with non-**UniProt** FASTA sequence databases as well as with file formats such as the PSI extended FASTA format (PEFF) that include known clinically relevant polymorphisms, is now underway.

## Conclusion

In metaproteomic analyses where search spaces can be vast, the accuracy of protein and taxonomic assignment is critical. It is important to note therefore that not only does **MetaNovo** identify a greater total number of proteins per microbiome sample than previous two step

approaches, but the clear separation of distributions of observed PEP scores between identified hits in the **MetaNovo**- and decoy sequence databases, coupled with a search space that does not artificially exclude host proteins, provides increased confidence in the spectral assignments made.

Benchmarking of **MetaNovo** against other two step search tools, including on samples of defined microbial composition, yielded favourable results, with **MetaNovo** providing increased sensitivity without overestimating taxa present in the samples whilst simultaneously identifying unexpected components that may be missed using limited or hand-selected sequence databases.

By estimating taxonomic and peptide level information on microbiome samples directly from tandem mass spectrometry data, in combination with the entire **UniProt** sequence database, **MetaNovo** allows simultaneous identification of human, bacterial, fungal, viral and other eukaryotic proteins in a given sample, unrestricted by prior expectation or availability of prior genomic data. The freely available MetaNovo algorithm should therefore enable clinically important correlations between changes in microbial protein abundance and change in the host proteome to be investigated in a single analysis, whilst minimising reliance on prior knowledge, or external information such as whole-genome sequencing.

## Supporting information

**S1 Table. Taxonomic distribution by Kingdom of the UniProtKB 2019_11 release.**
(XLSX)

**S2 Table. Organisms included in the 9MM samples.** Please refer to original publication, Table 1 [33].
(XLSX)

**S3 Table. MLI peptide PEP scores comparison.** Shared—Peptides identified in all runs. Only—Peptides only identified in a particular run. Reverse hits—Reverse peptide hits for a particular run.
(XLSX)

**S4 Table. MLI peptide PEP scores Dunn's test post hoc comparison.**
(XLSX)

**S5 Table. UniPept pept2lca analysis of MLI samples.**
(CSV)

**S6 Table. Peptide counts of taxa only identified with MetaNovo in the MLI samples.**
(XLSX)

**S7 Table. Peptide counts of taxa identified in all runs by UniPept *pept2lca* "taxon_name" in the MLI samples.**
(XLSX)

**S8 Table. "kingdom_name" peptides in the MLI samples.**
(XLSX)

**S9 Table. "phylum_name" peptides in the MLI samples.**
(XLSX)

**S10 Table. 9MM peptide PEP scores comparison.** Shared—Peptides identified in all runs. Only—Peptides only identified in a particular run. Reverse hits—Reverse peptide hits for a

particular run.
(XLSX)

**S11 Table. 9MM peptide PEP scores Dunn's test post hoc comparison.**
(XLSX)

**S12 Table. UniPept *pept2lca* analysis of 9MM samples.**
(CSV)

**S13 Table. 9MM *Gallus gallus* peptide BLAST (tabular).**
(XLSX)

**S14 Table. 9MM *Gallus gallus* protein BLAST (tabular).**
(XLSX)

**S1 Fig. Density plot of MLI peptide PEP scores by group.**
(TIF)

**S2 Fig. Bacteria "superkingdom_name" peptides in the MLI samples.**
(TIF)

**S1 Data. 9MM *Gallus gallus* peptide BLAST.**
(JSON)

**S2 Data. 9MM *Gallus gallus* protein BLAST.**
(JSON)

## Acknowledgments

We acknowledge the use of the ilifu cloud computing facility - www.ilifu.ac.za, a partnership between the University of Cape Town, the University of the Western Cape, the University of Stellenbosch, Sol Plaatje University, the Cape Peninsula University of Technology and the South African Radio Astronomy Observatory. The ilifu facility is supported by contributions from the Inter-University Institute for Data Intensive Astronomy (IDIA—a partnership between the University of Cape Town, the University of Pretoria and the University of the Western Cape), the Computational Biology division at UCT and the Data Intensive Research Initiative of South Africa (DIRISA).

## Author Contributions

**Conceptualization:** Matthys G. Potgieter, David L. Tabb, Jonathan M. Blackburn.

**Data curation:** Matthys G. Potgieter, Andrew J. M. Nel, Suereta Fortuin.

**Formal analysis:** Matthys G. Potgieter, Suereta Fortuin, Jerome M. Wendoh.

**Investigation:** Andrew J. M. Nel.

**Methodology:** Shaun Garnett, David L. Tabb, Jonathan M. Blackburn.

**Software:** Matthys G. Potgieter.

**Supervision:** David L. Tabb, Nicola J. Mulder.

**Writing – original draft:** Matthys G. Potgieter.

**Writing – review & editing:** Shaun Garnett, David L. Tabb, Nicola J. Mulder, Jonathan M. Blackburn.

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
