## [Decision Letter · Decision Letter 0]

7 Feb 2023

Dear Potgieter,

Thank you very much for submitting your manuscript "MetaNovo : an open-source pipeline for probabilistic peptide discovery in complex metaproteomic datasets" for consideration at PLOS Computational Biology. As with all papers reviewed by the journal, your manuscript was reviewed by members of the editorial board and by several independent reviewers. The reviewers appreciated the attention to an important topic. Based on the reviews, we are likely to accept this manuscript for publication, providing that you modify the manuscript according to the review recommendations.

The reviewers all agree on the high quality of the manuscript and raise only matters related to presentation. Thus, we expect that these matters can be addressed quickly.

Sincerely,

Luis Pedro Coelho

Academic Editor

PLOS Computational Biology

Lucy Houghton

Staff

PLOS Computational Biology

The reviewers all agree on the high quality of the manuscript and raise only matters related to presentation. Thus, we expect that these matters can be addressed quickly.

Reviewer's Responses to Questions

**Comments to the Authors:**

Reviewer #1: In this manuscript, Potgieter et al developed a metaproteomic data analysis tool, MetaNovo, which involves de novo sequencing in the first step for the refined database generation. MetaNovo enables peptide identification from a comprehensive Uniprot database directly. This method is widely applicable to a variety of multi-species samples. The peptides from the host and other contaminants can also be identified simultaneously, which can improve the identification rate and reduce the random matches. This work can help researchers in various fields aiming to study the proteome from multi-species samples. There are some minor concerns that need to be addressed before publication.

1. The users care about how fast the software runs. Could the authors provide the running time for the generation of the refined database from the Uniprot database by the de novo strategy?

2. Could the authors double-check the data used in Figure 3D and Figure 4D? It looks a little bit strange that there are parts of PSMs with higher PEP scores than the reverse parts. We know that the reverse parts represent the random matches, and if the “metaproiq-metagenome: only” in Figure 3D and the “Meta-PA: only” in Figure 4D are also random matches, the PEP distributions will be similar. However, these two parts showed significantly higher PEP scores than random matches. Besides, this trend was observed from two different datasets and was surprisingly consistent. If it’s correct, can you explore the reasons for this result?

3. Page 7, line 160. “for example using 16S sequencing data to produce a focused metaproteome library to search”. Here the word “library ” may cause a misunderstanding that it means a “spectra library”. I think it’s better to replace it with “database”.

4. Page 24, line 526-527. “The results were almost double those of the metagenome DB (Meta-PA) database that yielded 23.03 % identified MSMS, 6044 target peptides and 1087 target proteins”. What’s the size of the Meta-PA database? Is the size of Meta-PA too big which leads to lower identification rate? How many peptides can be identified if you use the MetaNovo workflow based on the Meta-PA database?

5. Page 26, line 551. “or UniPept misassignments of natural variants in the data.” The database used here is a Uniprot database, which incorporates taxonomic information. Are the identified protein names the same as the UniPept results? Actually, considering the Uniprot database contains the information of the organism for each protein, why use UniPept in the workflow instead of using the taxonomic information directly?

6. Page 33, line 714-716. The ethics statement mentioned that adult stool samples were used, but no related content was found in the article. Only two public datasets were downloaded and analyzed in the manuscript.

Reviewer #2: I think this paper contributes an important methodology to the proteomics and metaproteomics fields of study. It proposes that instead of making assumptions about the composition of organisms in a metaproteomics sample, we can use the vast amount of sequenced proteins that are publicly available and let the mass spectrometry data speak for itself. They do this by using de novo sequencing of peptides, followed by tag matching to the massive database of UniProt, and then they use their scoring and ranking metric to determine the which proteins should be put in the targeted database. Next, they take advantage of the standard database search of the mass spectrometry data, followed by using UniPept to determine what organism were present in the sample.

While a 2% increase in the number of identified MS/MS spectra is not astonishing, the fact that this pipeline allows the user to not make any assumptions on the composition of their sample is immensely useful. This research is of high importance to many different areas of research where metagenomics and metaproteomics are used as tools to study complex biological systems. MetaNovo is particularly applicable if a lab does not have to budget or expertise to do metagenomics sequencing in addition to metaproteomics. The authors showed that they were able to detect chicken egg yoke proteins from the media that their sample was cultured on. This indicates that the MetaNovo method will also provide a better list of contaminate proteins.

I was able to access their GitHub code and their PRIDE data. While their code base is not thoroughly documented, it is well organized and readable, and their GitHub page has instructions for how to install and run the MetaNovo software. Between their supplemental material, raw data on PRIDE, and their public GitHub code I believe that this study is repeatable by the scientific community.

The authors come to reasonable conclusions that are backed up by their data, both in their figures and supplemental material. Just from my own curiosity, I’d be interested in seeing approximately how much time the MetaNovo pipeline takes to run on three replicate MGF files on a personal workstation or on a cluster. Finally, line 760 says ‘S8_table.csv’ but the file downloaded is S8_figure.png.

Reviewer #3: This paper describes the algorithms, implementation, and results of a bioinformatics tool that uses de novo peptide identification to analyze LC-MS/MS metaproteomics data, specifically to create a database of protein sequences to use with traditional proteomics database search. The results show that the new tool has some advantages over more traditional methods of creating databases for metaproteomics/microbial community samples. The methods are sound and original, the results are interesting and the paper should be published. However, there are a few revisions that the authors could make to make their paper easier to understand. The authors should consider the following points:

1. Figures. Graphics resolution is low and type is very small in figures 3, 4, and 5. The small type is extremely hard to read and presents a real obstacle to understanding the paper. Authors should consider reformatting the figures, or breaking up multipart figures into multiple figures, to allow a larger font size. Possibly this is partly due to the automatic formatting in to a PDF file for reviewers, but as provided to the reviewer, the font size is too small. In addition, many plots are missing y-axis labels. I also found Figure 2 to be difficult to understand. What does the arrow represent? Do the colors encode important information? Authors should consider revising the graphic, significantly expanding the caption, or possibly both. Also consider making the y-axis of Figure 3D and 4D logarithmic to should differences at lot PEP values better. Captions should include the dataset used.

2. Dangers of omitting present organisms from the search database (page 8, lines 184-191). The authors cite Knudsen and Chalkley (reference 9 in the manuscript) as an example of what can happen when a true component of the sample is omitted from the search database. In this paper, Knudsen and Chalkley reexamine the data of Bromenshenk et al., who looked for protein markers of honeybee pathogens without including the honeybee protein database in their database. Knudsen and Chalkley do reach the conclusion that this omission was the cause of incorrect peptide identifications. However, Noble (https://www.nature.com/articles/nmeth.3450) also reexamined the Bromenshenk data and reached a different conclusion: that only a small number of spectral assigned to pathogens should have been assigned to honeybee peptides, and “the choice of database does not explain the incorrect results obtained by Bromenshenk et al.” (Noble). Since the authors invoke Knudsen and Chalkley’s conclusion as a reason for the increased success of MetaNovo (page 28 line 603), I would be very interested to hear how they account for Noble’s arguments in this context.

3. What do you mean by database? When describing each data set analyzed, the authors indicate the size of the search database produced by MetaNovo (e.g. page 17 lines 374-376 for the mucosal-luminal interface dataset). If I am following correctly, this “database” is the product of the whole MetaNovo pipeline, the “parsimonious protein subset identified” indicated near the bottom center of Figure 1. There is some possibility for the reader to confuse this database with the SQLite database that holds all the the DirecTag results, described on page 13 lines 287-291. The authors should consider how to distinguish these two uses of the word “database” to ensure that the results section is clear. I’m not just trying to nitpick here, it actually confused me the first time through.

4. Page 23 line 501. Clarify what is meant by “taxon_name taxa.”

**Have the authors made all data and (if applicable) computational code underlying the findings in their manuscript fully available?**

Reviewer #1: Yes

Reviewer #2: Yes

Reviewer #3: Yes

PLOS authors have the option to publish the peer review history of their article (what does this mean?). If published, this will include your full peer review and any attached files.

Reviewer #1: No

Reviewer #2: No

Reviewer #3: No

Figure Files:

Data Requirements:

Reproducibility:

References:

---

## [Editor Report · Decision Letter 1]

8 May 2023

Dear Prof Blackburn,

We are pleased to inform you that your manuscript 'MetaNovo : an open-source pipeline for probabilistic peptide discovery in complex metaproteomic datasets' has been provisionally accepted for publication in PLOS Computational Biology.

Best regards,

Luis Pedro Coelho

Academic Editor

PLOS Computational Biology

Ilya Ioshikhes

Section Editor

PLOS Computational Biology

Lucy Houghton

Staff

PLOS Computational Biology

---

## [Editor Report · Acceptance letter]

13 Jun 2023

PCOMPBIOL-D-22-01483R1 

MetaNovo: an open-source pipeline for probabilistic peptide discovery in complex metaproteomic datasets

Dear Dr Blackburn,

I am pleased to inform you that your manuscript has been formally accepted for publication in PLOS Computational Biology. Your manuscript is now with our production department and you will be notified of the publication date in due course.

With kind regards,

Anita Estes
